# Should social disconnectedness be included in primary-care screening for cardiometabolic disease? A systematic review of the relationship between everyday stress, social connectedness, and allostatic load

Anders Larrabee Sonderlund [ID]*☯, Trine Thilsing☯, Jens Sondergaard☯

The Research Unit for General Practice, Department of Public Health, University of Southern Denmark, Copenhagen, Denmark

☯ These authors contributed equally to this work.
* asonderlund@health.sdu.dk

## Abstract

In the present review, we argue that social disconnectedness could and should be included in primary-care screening protocols for the detection of cardiometabolic disease. Empirical evidence indicates that weak social connectedness represents a serious risk factor for chronic diseases, including cardiovascular disease, diabetes, and various cancers. Weak social connectedness, however, is largely regarded as a second-tier health-risk factor in clinical and research settings. This may be because the mechanisms by which this factor impacts on physical health are poorly understood. Budding research, however, advances the idea that social connectedness buffers against stress-related allostatic load–a known precursor for cardiovascular disease and cancer. The present paper reviews the empirical knowledge on the relationship between everyday stress, social connectedness, and allostatic load. Of 6022 articles retained in the literature search, 20 met predefined inclusion criteria. These studies overwhelmingly support the notion that social connectedness correlates negatively with allostatic load. Several moderators of this relationship were also identified, including gender, social status, and quality of social ties. More research into these factors, however, is warranted to conclusively determine their significance. The current evidence strongly indicates that the more socially connected individuals are, the less likely they are to experience chronic stress and associated allostatic load. The negative association between social connectedness and various chronic diseases can thus, at least partially, be explained by the buffering qualities of social connectedness against allostatic load. We argue that assessing social connectedness in clinical and epidemiological settings may therefore represent a considerable asset in terms of prevention and intervention.

**Data Availability Statement:** As this is a systematic review, all relevant data are within the manuscript and its Supporting Information files.

**Funding:** Outside of the University of Southern Denmark, the authors received no specific funding for this work.

**Competing interests:** The authors have declared that no competing interests exist.

# Introduction

## Background

Cardiometabolic diseases (CMD) such as cardiovascular disease (CVD), type 2 diabetes, and chronic obstructive pulmonary disease are the leading cause of premature death globally. In 2015, 17.7 million people died from CMD, including most predominantly coronary heart disease (7.4 million) and stroke (6.7 million) [1]. A plethora of empirical studies have shown that the majority of CMDs can be prevented by addressing behavioral risk factors such as unhealthy diet, tobacco and alcohol use, and leading a sedentary lifestyle (collectively the Big 4) [2]. Most research into the prevention of CMD to date has thus focused on changing behavior as it relates to the Big 4. Recent studies, however, indicate that there are other equally important modifiable risk factors beyond the Big 4, which are often overlooked in both research and practice. In particular, studies have emphasized the significance of social disconnectedness as a central determinant of CMD. Social connectedness refers to the actual and perceived nature and extent of the individual's ties to family and friends, social groups and networks [3]. In a noteworthy metanalysis of 148 studies (approximate collective N = 300.000), Holt-Lundstadt, Smith, and Layton [4] found that the quality and quantity of individuals' social relationships predicted morbidity and mortality above and beyond the Big 4. Specifically, the more socially disconnected people were, the more likely they were to develop CMDs, cancer, and mental health problems. The study accounted for age, gender, pre-existing health conditions, and SES. Lundstadt et al. [4] also argued that in spite of the evidence, weak social connectedness was grossly underestimated as a health-risk factor by health professionals and laypeople alike, and as such was seldomly addressed in healthcare settings. More recent studies indicate that little, if anything, has changed in this regard in the past decade [5, 6].

The discrepancy between knowledge and perception of weak social connectedness as a significant health-risk factor may reside in the notion that this construct–as it relates to physical health–is less intuitive in theory and practice compared to biological or behavioral risk factors. Haslam, Haslam, Jetten, Cruwys, and Bentley [7] also argue that the biopsychosocial model of health is overly reliant on biological health processes and tends to integrate psychological and social health determinants as secondary factors. Nonetheless, budding research has attempted to clarify the link between social connectedness and chronic somatic disease to improve the applicability of this knowledge in health care (e.g. in improving clinical risk assessment, CMD-preventive guidelines, and population-health initiatives) [5].

In theory, the mechanisms that underpin the relationship between social connectedness and physical health relate primarily to three distinct hypotheses: 1) social health norms that exist within social networks influence individual health behavior for better or for worse; 2) being embedded in a social network facilitates overall well-being, including physical health, through the availability of instrumental social support (e.g. financial support, work-life balance, health care); 3) being connected to others facilitates access to emotional and moral social support ("being there" for someone in times of need) which fosters individual psychological capital and resilience to everyday stress and the associated physiological wear and tear [8–10]. Thus, in terms of physical-health consequences, the former two hypotheses focus on the impact of social connectedness on CMD-risk factors (i.e. health norms, behavior, availability of concrete support), while the latter one centers more on biological endpoints (i.e. stress-related physiological reactivity that predicts CMDs). While these three mechanisms most certainly overlap to some extent (e.g. stress may lead to physiological reactivity *as well as* a propensity to engage in health-risk behaviors such as smoking), this review deals exclusively with the third pathway that focuses on the stress-buffering capacity of social connectedness and consequent physiological outcomes. We focus specifically on the toll of *everyday stress* (e.g.

work-related stress, financial stress, social adversity) as opposed to acute, temporary stressors (e.g. the death of a loved one).

Strong scientific evidence shows a positive association between CMD and various types of everyday stress, including work-related stress [11–13], stress related to financial adversity [14, 15], marital stress [15], caregiving [15], and the regular experience of discrimination [16–20]. These associations remain strong even when controlling for demographic, social, and health-behavioral factors [21, 22, 23]. Emerging research has increasingly focused on *allostatic load* as the key physiological manifestation of stress that over time may lead to CMD [24–29]. The term allostatic load (AL) was coined by McEwen and Stellar [30] who defined it as the cumulative physiological wear and tear that results from the regular experience of psychological stress. Specifically, hypothalamic-pituitary-adrenal (HPA) axis reactivity naturally fluctuates in response to external stressors–a phenomenon referred to as *allostasis*. In a healthy setting with no lingering stressors, the individual's physiological system will oscillate naturally between allostasis and the baseline level of homeostasis as stress arises and abates. However, when stress becomes chronic, so too does allostasis persist over time, taxing the physiological system and ultimately resulting in AL [30]. Consistent with the evidence base on stress and CMD, the specific link between AL and serious illnesses–including CMDs and cancer–has been supported in a number of previous studies and reviews [8, 9, 13, 16, 17, 22, 31–33], advancing the idea that AL is a central mediator of the physical health consequences of everyday psychological stress.

Combining the evidence on the relationship between social connectedness and physical health with the findings on stress, AL, and CMD, we argue that a key mechanism by which social connectedness protects individual health is by effectively buffering against AL. Indeed, since the early 2000s, studies explicitly testing this hypothesis have slowly, but steadily gained momentum and credibility. This research, however, spans decades and multiple methodologies and research designs, generating a somewhat disjointed empirical narrative. Thus, the aim of the present paper is to consolidate and synthesize the current evidence base on the association between social connectedness, stress, and AL.

## Rationale

To the authors' knowledge, few other reviews in this area have been conducted, and most (if not all) have focused on the broader relationship between for instance social support and cardiovascular markers of health (i.e. not necessarily stress-related reactivity). The gradual introduction of AL into the mainstream literature, however, has prompted a conceptual shift in the field by operationalizing stress as a psychophysiological phenomenon rather than a purely psychological one. Uchino et al. [9] conducted a highly informative review of the relationship between social support and physiological processes, parts of which focused on the impact of acute stress (not everyday stress) on physiological reactivity. Ten years on, Uchino [8] published a narrative review on this topic, again with sections on social support and physiological acute-stress reactions. More recently, Hostinar [34] and Wiley, Bei, Bower, and Stanton [35] also conducted reviews in this area. However, the former examines only literature published between 2013 and 2015, and the latter focuses on the relationship between AL and broad psychosocial measures rather than social connectedness per se. Thus, while we have identified four relevant reviews, two are 13 and 23 years old [8, 9], and none have explicitly focused on the link between social connectedness, everyday stress, and AL. In light of this, by critically assessing the entire body of research on this topic, the authors hope to make a significant addition to the field by updating and specifying past review efforts. Specifically, we will identify any relevant themes and variations that might have been missed in past reviews or appeared in

subsequent research, qualitatively assess the evidence base, and highlight avenues for further study. Ultimately and most importantly, we believe that mapping out the link between social connectedness, stress, and CMD will help crystallize the health-protective mechanisms of social connectedness, and push it to the fore in clinical as well as research settings so it may be appropriately regarded as a crucial health-risk factor. Specifically, we argue that the integration of social connectedness measures in clinical health-assessment algorithms will improve algorithm accuracy and usability.

## Method

### Protocol

This review was conducted according to the Preferred Reporting Items for Systematic Reviews and Meta-Analyses (PRISMA) guidelines. Full details can be accessed at www.prisma-guidelines.org.

### Literature search strategy

A diverse range of scientific journals for literature on the relationship between social connectedness and AL was examined. Specifically, a comprehensive and rigorous search of the following EBSCOhost databases was executed: Academic Search Premier; AMED; Global Health; SocINDEX with Full Text; CINAHL Plus with Full Text; E-Journals; MEDLINE; Psychology and Behavioral Sciences Collection; SCOPUS; Science Direct. A literature search using Web of Knowledge and Google Scholar to identify any additional references that might have been missed was also performed. Additionally, reference lists of relevant papers were manually searched.

The predefined Boolean/phrase search terms related directly to the conceptualization of the key variables of interest: Social connectedness and allostatic load. For the sake of comprehensiveness, social connectedness was defined as an umbrella term, covering measures of number and/or quality of individual social ties, actual and perceived group memberships, and extent of social network. Measures of social support were also included as a proxy for social connectedness. In terms of AL, articles that dealt explicitly with this type of physiological reactivity represented the main focus. However, research that might not refer specifically to AL, but which nonetheless used AL biomarkers as stress-related outcomes was also included. For these latter studies, the Allostatic Load Index (ALI) was consulted to assess relevance. ALI is a commonly accepted measure of AL that comprises neuroendocrine and cardiovascular biomarkers, including systolic and diastolic blood pressure, total cholesterol, high-density-lipoprotein, glycosylated hemoglobin, waist-to-hip ratio, dehydroepiandrosterone sulfate, urinary epinephrine, norepinephrine, cortisol, as well as inflammatory markers such as C-reactive protein and cytokine levels [36].

The exact literature search syntax was as follows: "Allostasis" AND/OR "Allostatic load" AND/OR "Physiological reactivity" AND/OR "Psychophysiological reactivity" AND "Social connectedness" AND/OR "Group membership" AND/OR "Social identity" AND/OR "Social ties" AND/OR "Social network". Search limiters were also specified to exclude obviously unrelated topics, such as for example papers on professional practice, philosophy, case studies, etc. Once the search had been executed, papers were selected based on the following inclusion criteria:

1. The paper reported empirical studies on the relationship between human social connectedness, chronic and/or everyday (not acute) chronic stress, and AL.

2. The full text was available.

3. The paper was in English.

4. The paper had undergone scientific peer review.

5. The paper had been published since 1990.

Each database hit was evaluated by the authors in three rounds against the inclusion criteria. In the first round, papers that clearly did not relate to the subject matter were rejected (usually based on title). In the second round, abstracts of the papers retained in the first round were reviewed. Any article that failed to meet the inclusion criteria was discarded. Finally, the papers that remained after the first two evaluation rounds were downloaded and scrutinized in full-text detail for relevance. Only papers that passed through each of these three rounds were ultimately retained for inclusion in the review.

### Research quality appraisal

The final round of search results evaluation included an appraisal of the retained papers' methodological quality. To this end, the Quality Assessment Tool for Quantitative Studies (QATQS) was employed [37]. The QATQS evaluates research along six dimensions, including study population selection bias, study design, confounding variables, researcher blinding, data collection methods, and participant withdrawal and attrition. Each dimensional score for a given paper is averaged and combined into an overall assessment of the paper in terms of 'weak', 'moderate', or 'strong'. The primary and secondary authors each coded all of the papers independently, and any differences in assessment were resolved through discussion and re-examination of the given paper.

Due to the small number of studies retained ($n = 20$), a meaningful relative weighting by study effect sizes (meta-analysis) was not feasible. Instead the strengths and weaknesses of each study were taken into account in a synthesis of overall findings in the discussion section.

## Results

### Literature search results

The initial search identified a total of 6022 articles. Of these, a subset of 144 papers were identified that in some way dealt with the convergence of social cognition/environment and physiological reactivity. Most of these articles were rejected due to one or more of the following reasons: The paper did not report on social connectedness as an independent, mediator, or moderator variable; the paper used composite measures of their social and physiological factors, preventing any isolated assessment of the variables of interest; the paper did not cite empirical research (e.g. editorial, comment); the paper reported insufficient statistical and/or methodological detail for assessment; the paper reported on animal populations; or a combination of these issues (see Fig 1). Ultimately, a total of 20 papers for were included in the present review. In the following sections, this evidence will be critically reviewed as it relates to the AL-buffering qualities of social connectedness.

### Study characteristics and methodology

Of the 20 papers included in this review, 12 were from the 2010s, and the remaining eight articles were published in the 2000s. Further, the vast majority of articles was US-based ($n = 17$), with only one other nationality represented (Taiwan). The study populations varied somewhat in terms of ethnicity (African-Americans, Mexican-Americans, White Americans, Taiwanese) and age (teenagers to 100-year-olds). The average sample size across studies was N = 1024 (min = 54, max = 6729). In terms of methodology, 10 studies reported cross-sectional data, six

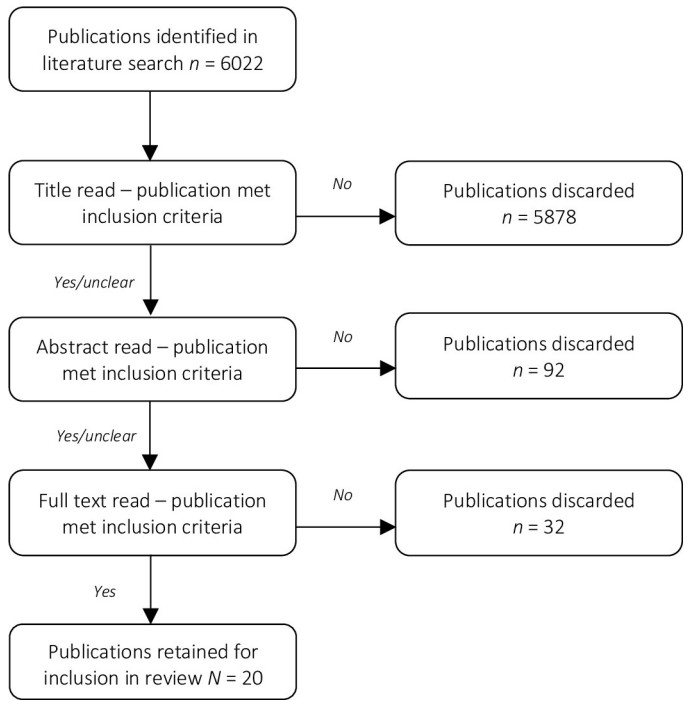

**Fig 1. Article evaluation process flow chart.**

were longitudinal studies, three were multi-method longitudinal/cross-sectional studies, and one was quasi-experimental. The average timeframe for longitudinal studies was about nine years (min = 2 years, max = 18 years, median = 10 years).

The social connectedness and AL measures employed varied somewhat across studies. Social connectedness was assessed by individual or composite measures of perceived social connectedness, social isolation, social support, social network size, group membership, social contact, social integration, and/or social ties. Beyond the sheer extent and nature of social connectedness, many studies also included measures of connection quality–for example in terms of relationship negativity or social network strain. When measuring established constructs such as social support or social network size and/or composition, 15 studies used validated scales (e.g. the Berkman-Syme Social Network Index), however five articles either did not report which scales had been employed, or constructed scales specifically for their studies. In terms of AL, 15 studies used either the ALI or a composite measure of AL that included many if not all of the ALI biomarkers. On the other hand, some studies focused on a select few AL indicators, such as C-reactive protein (CRP), fibrinogen, serum albumin levels [38, 39] (see Table 1).

## Research methodology and quality

The inter-rater research quality assessments, conducted by the primary and secondary authors, were overall well-aligned. In the initial round of assessments, assessments deviated on only one qualitative dimension in each of three papers (81.25% inter-rater reliability). These discrepancies were resolved by reexamination of the given papers and subsequent discussion. Ultimately, scoring each paper along the six qualitative dimensions designated by the QATQS, 13 papers were deemed as based on 'strong' research, four as 'moderate', and three as 'weak'

**Table 1.** *Study characteristics.*

| Author | Country | Population (*N*) | Research Design | Predictor variable (constructs/scales) | Covariates | Allostatic Load (AL) markers | Findings | Study quality |
|---|---|---|---|---|---|---|---|---|
| Brody et al. (2014a) | USA | African-American 11-19-year-old youths. living in socially deprived areas (420) | Cross-sectional | • Changes in neighborhood deprivation. <br> • Emotional support (family, peer, adult mentor; composite of the *Family Support Inventory*, the *Carver Support Scale*, the *Mentor Relationship Index*) | • Gender <br> • Diet <br> • Smoking status <br> • Alcohol use <br> • Perceived life stress at age 19 <br> • SES | AL composite: <br> • BP <br> • BMI <br> • Catecholamine levels <br> • Cortisol levels <br> • Norepinephrine levels <br> • Epinephrine levels | Increasing neighborhood poverty levels between participant ages of 11 to 19 correlated positively with AL ($\beta$ = 0.21*). Emotional support moderated this association such that poverty correlated positively with AL in youths who had low levels of emotional support ($\beta$ = 0.61**). This association was non-significant for youths who received high emotional support ($\beta$ = 0.10, $p$ = .52). | Strong |
| Brody et al. (2014b) | USA | African-American adolescents (331) | Longitudinal (two yrs.) | • Perceived discrimination <br> • Emotional support (composite of family, peer; the *Family Support Inventory*) | • SES <br> • Perceived stress <br> • Depressive symptoms <br> • Diet <br> • Physical activity <br> • Smoking status <br> • Alcohol use <br> • Marijuana use | AL composite: <br> • BP <br> • BMI <br> • C-reactive protein <br> • Cortisol levels <br> • Norepinephrine levels <br> • Epinephrine levels | Results indicated a significant positive relationship between high and stable level of perceived discrimination and AL (B = 1.09*). This association remained true for participants who had low emotional support (B = -1.45**), but not for participants who received high emotional support. | Strong |
| Brooks et al. (2014) | USA | National sample of 34-84-year-old adults (949) | Longitudinal (10 yrs.) | • Emotional support (family, friends, spouse/partner) <br> • Social negativity (family, friends, spouse/partner) <br> • Social contact (family, friends) | • Age <br> • Gender <br> • Race <br> • Education <br> • Smoking status <br> • Physical activity <br> • Major health conditions <br> • Disabilities <br> • Mental health | AL composite: <br> • Systolic BP <br> • Pulse pressure <br> • Resting pulse rate <br> • Epinephrine levels <br> • Norepinephrine levels <br> • Heart-rate variability <br> • Hormone cortisol levels <br> • DHEAS levels <br> • C-reactive protein levels <br> • Fibrinogen levels <br> • Interleukin-6levels <br> • E-selectin molecule <br> • Intracellular adhesions molecule-1 <br> • HDL/LDL cholesterol levels <br> • Triglycerides levels <br> • BMI <br> • Waist-hip ratio <br> • Glycosylated hemoglobin levels <br> • Fasting glucose levels <br> • Insulin resistance | Higher levels of spouse negativity ($\beta$ = 0.16*) and family negativity ($\beta$ = 0.14*) were positively correlated with AL. Higher levels of spouse support were negatively associated with AL ($\beta$ = -0.19*). Friend ($\beta$ = 0.01*) and SN support ($\beta$ = 0.02*) were associated with higher AL among older adults. For younger adults, there was no association between friends support and AL, and significant negative associations between SN support and AL ($\beta$ = -0.02*). SN negativity was positively associated with AL among younger adults only ($\beta$ = -0.01*). | Strong |

*(Continued)*

**Table 1.** (Continued)

| Author | Country | Population (*N*) | Research Design | Predictor variable (constructs/scales) | Covariates | Allostatic Load (AL) markers | Findings | Study quality |
|---|---|---|---|---|---|---|---|---|
| Friedman et al., (2015) | USA | National representative sample of healthy 25-74-year-olds (1180) | Cross-sectional | • Early-life SES adversity<br>• Childhood adversity (abuse, parental divorce or death) | • Race<br>• Age<br>• Sex<br>• Education<br>• Social relationships (strained vs. supportive)<br>• Smoking status<br>• Alcohol consumption<br>• Physical activity | AL composite<br>• Systolic BP<br>• Systolic BP<br>• Pulse pressure<br>• Resting pulse rate<br>• Epinephrine levels<br>• Norepinephrine levels<br>• Hormone cortisol levels<br>• DHEAS levels<br>• C-reactive protein levels<br>• Fibrinogen levels<br>• Interleukin-6levels<br>• E-selectin molecule<br>• Intracellular adhesions molecule-1<br>• Triglycerides levels<br>• BMI<br>• Waist-hip ratio<br>• Glycosylated hemoglobin levels<br>• Fasting glucose levels | Results indicated a dose-response relationship between early-life adversity and AL where AL increased by 0.093 for each additional adverse experience. This effect was moderated by social relationships such that social strain exacerbated the impact on AL, while social support assuaged it. Specifically, social relationships accounted for 19% of the adversity-AL association. This moderation effect, however, was statistically non-significant. | Strong |
| Gersten (2008) | Taiwan | Nationally representative sample of Taiwanese >50 years old (880) | Cross-sectional | • Number and frequency of stressors (familial stress, financial situation, employment, marital stress) | • Social connectedness (marital status, cohabitation, group membership)<br>• Age<br>• Sex<br>• Education<br>• Urban residence<br>• Mainlander ethnicity<br>• Diet<br>• Physical activity<br>• Smoking status<br>• Medication use | AL composite:<br>• Systolic BP<br>• Diastolic BP<br>• Epinephrine levels<br>• Norepinephrine levels<br>• Dopamine levels<br>• Hormone cortisol levels<br>• DHEAS levels<br>• Interleukin-6levels<br>• HDL/LDL cholesterol levels<br>• Triglycerides levels<br>• BMI<br>• Waist-hip ratio<br>• Glycosylated hemoglobin levels<br>• Fasting glucose levels | Results indicated no statistically significant correlation between lifetime stress and AL. There was, however, a positive association between current stress and AL. Results pertaining to the moderating effect of social connectedness were inconclusive. In one regression model, social connectedness interacted with number of stressors experienced, while in another model it interacted with stress frequency only. | Strong |

(*Continued*)

**Table 1.** (Continued)

| Author | Country | Population (N) | Research Design | Predictor variable (constructs/scales) | Covariates | Allostatic Load (AL) markers | Findings | Study quality |
|---|---|---|---|---|---|---|---|---|
| Glei et al. (2007) | Taiwan | Nationally representative sample of Taiwanese >50 years old (916) | Cross-sectional/ longitudinal | • Number of chronic stressors (e.g. marital stress, moving, health issues, financial stress) | • Perceived stress<br>• Social network (size, contact, activity, emotional support)<br>• Position in social hierarchy (SES, education)<br>• Internal resources (locus of control, engagement in everyday tasks, optimism)<br>• Age<br>• Sex<br>• Urban residence | AL composite:<br>• Systolic BP<br>• Diastolic BP<br>• Epinephrine levels<br>• Norepinephrine levels<br>• Dopamine levels<br>• Hormone cortisol levels<br>• DHEAS levels<br>• Interleukin-6levels<br>• HDL/LDL cholesterol levels<br>• Triglycerides levels<br>• BMI<br>• Waist-hip ratio<br>• Glycosylated hemoglobin levels<br>• Fasting glucose levels | The study reports a positive relationship between number of stressors and AL. Perceived stress did not mediate this effect. The combination of low social position, weak social networks, and limited internal resources rendered individuals more vulnerable to AL, though effect sizes were small and non-significant. | Strong |
| Gruenewald et al. (2012) | USA | 35-85-year-olds (1008) | Cross-sectional | • SES adversity in childhood (financial stress, parental education, childhood welfare status) and adulthood (education level, family-size adjusted income to poverty ratio, current financial situation, availability of money for basic needs, difficulty paying bills | • Age<br>• Sex<br>• Race<br>• Health conditions<br>• Alcohol consumption<br>• Smoking status<br>• Physical activity<br>• Perceived stress (*the Perceived Stress Scale*)<br>• Depression (*Center for Epidemiologic Studies Depression Scale*)<br>• Anxiety (*Mood and Symptoms Quesionnaire*)<br>• Perceived mastery and constraints<br>• Social contact<br>• Social support<br>• Social conflict | AL composite:<br>• Systolic BP<br>• Diastolic BP<br>• Resting heart rate<br>• BMI<br>• Waist-hip ratio<br>• Triglycerides levels<br>• HDL/LDL cholesterol levels<br>• Glycosylated hemoglobin<br>• Fasting glucose levels<br>• Insulin resistance<br>• C-reactive Protein levels<br>• IL6 levels<br>• Fibrinogen levels<br>• sE-Selectin levels<br>• sICAM-1 levels<br>• Epinephrine levels<br>• Norepinephrine levels<br>• Cortisol levels<br>• DHEA-S levels<br>• SDRR<br>• RMSSD<br>• Low/high frequency spectral power | Results indicated a positive relationship between total SES adversity and AL. This relationship was non-significantly moderated by light alcohol consumption and frequency of contact with friends. | Strong |
| Hawkley et al. (2011) | USA | Population-based sample of 51–69 year old White, Black, and Hispanic adults (208) | Cross-sectional | • SES (Education, Household income)<br>• Social support (emotional, instrumental; *Interpersonal Support Evaluation List*)<br>• Social network (social identities, church attendance, co-habitation, overall social network; adapted *Berkman-Syme Social Network Index*) | • Chronic health conditions<br>• Stress<br>• Diet<br>• Physical activity<br>• Alcohol use<br>• Smoking status<br>• Sleep pattern | AL composite:<br>• Systolic BP<br>• Diastolic BP<br>• HDL cholesterol<br>• Total cholesterol<br>• Glycosylated hemoglobin levels<br>• Cortisol excretion levels<br>• DHEAS levels<br>• Norepinephrine levels<br>• Epinephrine levels | Results indicated null relationships between AL and both social support and network indices. However, these variables were probed purely as mediators. | Weak |

(*Continued*)

**Table 1.** (Continued)

| Author | Country | Population (N) | Research Design | Predictor variable (constructs/scales) | Covariates | Allostatic Load (AL) markers | Findings | Study quality |
|---|---|---|---|---|---|---|---|---|
| Maselko et al. (2007) | USA | 70-80-year-old residents in NC, MA, CT (853) | Cross-sectional | • Religious activity (church attendance) | • Age<br>• Gender<br>• Ethnicity<br>• Income<br>• Education<br>• Marital status<br>• Physical functioning<br>• Presence of coronary heart disease, diabetes, or cancer<br>• Social ties (number of ties and frequency of contact)<br>• Social support (quality of emotional and instrumental support) | AL composite:<br>• Systolic BP<br>• Diastolic BP<br>• HDL cholesterol<br>• Total cholesterol<br>• Glycosylated hemoglobin levels<br>• Cortisol excretion levels<br>• DHEAS levels<br>• Norepinephrine levels<br>• Epinephrine levels | Religious service attendance (min. once a week) was negatively correlated with overall AL in women ($\beta$ = -0.47**), but not in men. Weekly religious service attendance was associated with a 61% decreased risk of having high AL (OR = 0.39, 95% CI 0.20, 0.76) for women, but not for men. Social integration had no discernible impact on this relationship. | Moderate |
| McClure et al. (2015) | USA | US-based Mexican immigrant adults (126) | Cross-sectional | • Social support (emotional, instrumental) | • Age<br>• Alcohol use<br>• TV<br>• Pain level | AL composite:<br>• C-Reactive Protein levels<br>• Systolic BP<br>• Diastolic BP<br>• Fasting glucose levels<br>• Total cholesterol<br>• Waist-hip ratio | Among women, family support was negatively associated with AL (OR = 8.23, 95% CI 2.06, 32.92), but only in majority White communities as opposed to Mexican enclaves. | Weak |
| Miller et al. (2002) | USA | Parents of chronically ill children (25), parents of healthy children (25) | Quasi-experimental | • Social support (emotional, instrumental) (*the Interpersonal Support Evaluation List* mod)<br>• Perceived stress (*the Perceived Stress Scale*)<br>• Mood (*Profile of Mood States*)<br>• Depression (*Center for Epidemiologic Studies–Depression Scale*) | • Smoking<br>• Alcohol consumption<br>• Physical activity<br>• Sleep quality | AL markers<br>• Cortisol levels (diurnal pattern, total day secretion)<br>• Glucocorticoid sensitivity<br>• Cytokine production (IL-6, TNF-$\alpha$, IL-1$\beta$) | Group membership and perceived instrumental social support interacted to impact negatively (i.e. resistance increased) on glucocorticoid sensitivity in PCICs (simple slope = -.05), but not PHCs (simple slope = .01). Instrumental social support thus appeared to buffer against stress and AL. | Weak |
| Rosal et al. (2004) | USA | 20-70-year-olds with chronic stress (146) | Cross-sectional/ longitudinal | • Social support (*the MOS Social Support scale*)<br>• Stress (*the Hassles Scale; the Life Events List* (mod); *the Levenstein Perceived Stress Questionnaire; the Beck Depression Inventory, the Beck Anxiety Inventory*) | • Gender<br>• Age<br>• Ethnicity<br>• Marital status<br>• Education<br>• Employment status<br>• Caffeine consumption<br>• Alcohol consumption<br>• Physical activity<br>• Smoking status<br>• BMI<br>• Hostility<br>• Mental health | AL markers<br>• Cortisol levels (morning, evening, daytime change) | Cross-sectional and longitudinal results were comparable and indicated an unexpected inverse association between chronic stress and morning and daytime cortisol levels ($\beta$ = -.23**, $\beta$ = -.22**). Social support also correlated negatively with stress ($\beta$ = -.53**). Social support was inversely associated with cortisol levels such that cortisol was 36% lower ($\beta$ = -.44**) for those in the upper tertile of social support vs. those in the two lower tertiles. Social support appeared to buffer against stress and associated cortisol fluctuation (AL). | Moderate |

(*Continued*)

**Table 1.** (Continued)

| Author | Country | Population (N) | Research Design | Predictor variable (constructs/scales) | Covariates | Allostatic Load (AL) markers | Findings | Study quality |
|---|---|---|---|---|---|---|---|---|
| Seeman et al. (2002) | USA | Community-based cohorts. Cohort 1: 70-79-year-olds (765), Cohort 2: 58-59-year-olds (106). | Cross-sectional cohort | • Parental ties (positive vs. negative; *Parental Bonding Scale*) <br> • Partner ties (emotional, sexual, intellectual, recreational; *Personal Assessment of Intimacy Relationships Inventory*, subscales) <br> • Social integration <br> • Social support (emotional, instrumental) | • Age <br> • Gender <br> • Education <br> • Ethnicity <br> • Marital status <br> • Income <br> • Self-rated health status | AL composite: <br> • Systolic BP <br> • Diastolic BP <br> • Waist/hip ratio <br> • HDL cholesterol <br> • Total cholesterol <br> • Glycosylated hemoglobin levels <br> • DHEA-S levels <br> • Cortisol excretion levels <br> • Norepinephrine levels <br> • Epinephrine levels | In the younger cohort, there was a significant negative association between positive cumulative relationship experiences and AL for men (OR = 0.25, 95% CI 0.08, 0.75) and women (OR = 0.22, 95% CI 0.06, 0.86). In the older cohort, there was a negative relationship for men between AL and both social integration ($\beta$ = -0.03*) and frequent emotional support ($\beta$ = -0.33*). Similar relationships were observed for women, but these were non-significant. Men were more likely to exhibit AL in terms of cardiovascular parameters, whereas women tended to exhibit AL in terms of neuroendocrine parameters. | Strong |
| Seeman et al. (2004) | Taiwan | Near-elderly (54–70) and elderly (71+) Taiwanese (1023) | Longitudinal (10 yrs.) | • Marital status <br> • Number of social ties <br> • Social activity <br> • Emotional support <br> • Criticism from others <br> • Excessive demands from others | • Gender <br> • Age <br> • Ethnicity <br> • Male respondent's education <br> • Financial strain <br> • Self-rated health <br> • Physical disability <br> • Spouse's health | AL composite: <br> • Systolic BP <br> • Diastolic BP <br> • Waist/hip ratio <br> • HDL cholesterol <br> • Total cholesterol <br> • Glycosylated hemoglobin levels <br> • DHEA-S levels <br> • Cortisol excretion levels <br> • Norepinephrine levels <br> • Epinephrine levels | Few ties with close friends/neighbors was positively correlated with AL ($\beta$ = 0.29**) for near-elderly men, but not for women. The perceived quality of social relationships was not consistently related to AL. Strained relationships were associated with higher AL ($\beta$ = 0.29**) in near-elderly. Low support was associated with low AL in near-elderly women only ($\beta$ = -0.63*). Among the elderly, number of ties with non-relatives was correlated negatively with AL ($\beta$ = 0.29**). | Strong |
| Seeman et al. (2014) | USA | 32-45-year-olds (844) | Longitudinal (15 yrs.) | • Social network <br> • Emotional support <br> • Social strain (non-supportive network) | • Age <br> • Sex <br> • Race <br> • Education <br> • Physical activity <br> • Smoking status | AL composite: <br> • Systolic BP <br> • Diastolic BP <br> • Heart-rate variability <br> • C-reactive protein <br> • Interleukin-6 levels <br> • Norepinephrine levels <br> • Epinephrine levels <br> • Waist circumference <br> • HDL/LDL cholesterol levels <br> • Triglycerides levels <br> • Glucose levels <br> • Insulin levels | Number of close social relationships (Cohen's $d$ = 0.22*) and frequency of received social support from close family and friends (Cohen's $d$ = 0.26*) correlated negatively with AL. This relationship is particularly apparent for inflammatory, metabolic, and autonomic risks. Frequency of social strain correlated positively with AL (Cohen's $d$ = 0.74*). Only social strain impacted significantly on AL when including all three variables in a statistical model simultaneously. | Strong |

(*Continued*)

**Table 1.** (Continued)

| Author | Country | Population (N) | Research Design | Predictor variable (constructs/scales) | Covariates | Allostatic Load (AL) markers | Findings | Study quality |
|---|---|---|---|---|---|---|---|---|
| Sotos-Prieto et al. (2015) | USA | 45-75-year-old Puerto Ricans living in Boston, MS (787). | Cross-sectional | • Diet<br>• Physical activity<br>• Sedentary behaviors<br>• Smoking status<br>• Social support (emotional, instrumental; *Norbeck Social Support Questionnaire*)<br>• Social network (size; *Norbeck Social Support Questionnaire*)<br>• Social integration (social activity; *Social and Community Support and Assistance Questionnaire*)<br>• Sleep pattern | • Age<br>• Sex<br>• Education<br>• Household income<br>• Medication use | AL composite:<br>• Systolic BP<br>• Diastolic BP<br>• DHEA-S levels<br>• Serum insulin levels<br>• Serum glucose levels<br>• Total cholesterol levels<br>• HDL/LDL cholesterol levels<br>• Plasma TG levels<br>• Glycated hemoglobin levels<br>• Cortisol levels<br>• Epinephrine/ Norepinephrine levels | Quality and size of social support and network correlated negatively with AL (Cohen's d = 0.24**). | Moderate |
| Weinstein et al. (2003) | USA/ Taiwan | 60-100-year-old Taiwanese (101) and Americans (827) | Longitudinal (seven yrs.) | • Position in social hierarchies (sex, education, income, occupation)<br>• Social network (extent of social activity; extent of weekly contact with friends/ neighbors; extent of contact with his/her children) | • Age<br>• Sex<br>• Education<br>• Income<br>• Occupation | AL composite:<br>• Systolic BP<br>• Diastolic BP<br>• Waist/hip ratio<br>• Urinary cortisol levels<br>• Urinary adrenaline and noradrenaline levels<br>• Glycosylated haemoglobin levels<br>• DHEA-S levels<br>• HDL cholesterol levels<br>• Total cholesterol levels | Social connectedness correlated negatively, but non-significantly with AL. Widowhood, however, was significantly and positively correlated with AL ($\beta$ = 0.24*). | Moderate |
| Yang et al. (2013) | USA | 60-year-olds and over (6729) | Longitudinal (18 yrs.) | • Social network (marital status, contacts with friends/ relatives, religious attendance, group membership; *Berkman-Syme Social Network Index*) | • Age<br>• Sex<br>• Race<br>• Education<br>• Family income<br>• Smoking status<br>• Alcohol use<br>• Physical activity<br>• BMI<br>• Medical history<br>• Self-rated health | AL markers<br>• C-reactive protein levels<br>• Fibronogen levels<br>• Serum albumin levels. | The extent and quality of social network correlated negatively with CVD by buffering against stress-related physiologic inflammation in both men (Hazard Ratio = 1.49**) and women (Hazard Ratio = 1.47**). Inflammation partially mediated this relationship, accounting for 12% of overall association between social connectedness and CVD (95% CI -0.35, -0.2). | Strong |
| Yang et al. (2014) | USA | Sample of current/ past cancer patients over 20 years old (1075) | Cross-sectional | • Social network (marital status, contacts with friends/ relatives, religious attendance, group membership; *Berkman-Syme Social Network Index*) | • Age<br>• Sex<br>• Race<br>• Education<br>• Family income<br>• Smoking status<br>• Alcohol use<br>• Physical activity<br>• BMI<br>• Medical history<br>• Self-rated health<br>• Cholesterol medication use | AL markers<br>• C-reactive protein levels<br>• Fibrinogen levels<br>• Serum albumin levels. | Social network size and quality was negatively associated with stress-related inflammation (SN bracket 1 (low): OR = 2.35, 95% CI 1.62, 3.40; SN bracket 2: OR = 1.69, 95% CI 1.21, 2.36; SN bracket 3: OR 1.49, 95% CI 1.08, 2.06; SN bracket 4 (high): OR = 1.00)**. The relationship resembled a dose-response relationship. | Strong |

*(Continued)*

**Table 1.** (Continued)

| Author | Country | Population (N) | Research Design | Predictor variable (constructs/scales) | Covariates | Allostatic Load (AL) markers | Findings | Study quality |
|---|---|---|---|---|---|---|---|---|
| Yang et al. (2015) | USA | Older adults aged 57–85 years old (1264) | Longitudinal (six yrs.) & Cross-sectional | • Social network (marital status, religious attendance, frequency of socializing and volunteering; *Berkman-Syme Social Network Index*) <br> • Social support (emotional, instrumental) | • Age <br> • Sex <br> • Race <br> • Education <br> • Hypertension medication use <br> • Psychosocial stressors <br> • Smoking status <br> • Physical activity <br> • Alcohol use <br> • BMI <br> • Diabetes | AL markers <br> • Systolic BP <br> • Hypertension diagnosis | Results from the cross-sectional analysis found that people with poor social networks were 65% more likely than people with high social integration to have hypertension (OR = 1.65*, 95% CI 0.99, 2.76). Results from the longitudinal analysis indicated that both social support and integration were inversely associated with perceived stress and both SBP and hypertension rates. However, here social support was the dominant factor in terms of SBP, whereas social integration accounted for most of the variance in hypertension rates. | Strong |

(see Table 1). The current evidence base on the relationship between social connectedness and AL thus rests on research of generally high methodological rigor.

## Study findings

In the following paragraphs, a brief description of the main findings reported in the retained articles will be provided (see Table 1 for detailed information about study samples, design, and methodology).

In two studies, Brody et al. [40, 41] found that emotional support–derived from family, peer, and mentor networks–buffered against AL associated with neighborhood poverty and discrimination stressors. This effect remained even when controlling for a host of potentially confounding variables, including gender, health behavior, and SES. These findings were further backed up in similar research by Sotos-Prieto et al. [42] and Rosal et al. [43] who found that the extent of the individual's social network and integration, and/or the quality of social support derived from their network, was inversely associated with AL. Indeed, the latter study found a 36% reduction in cortisol levels between those participants with the least amount social support versus those with the most.

A more extensive, ten-year longitudinal analysis of the association between social connectedness and AL, discovered a complex relationship that was moderated by several factors. These included age, the nature of ties with others (i.e. negative vs. positive), and the source of support received (spouse, friends, entire social network) [44]. Specifically, results indicated that negative and stressful relationships with a spouse and family members correlated positively and significantly with AL. Spousal support, on the other hand, correlated negatively with AL. Similarly, social network support correlated negatively with AL in young adults, while social network negativity correlated positively with AL. Counterintuitively, friend support and social network support were positively associated with AL in older adults, though the authors speculated that this might be due to greater feelings of social obligation in this subpopulation.

In another study, McClure et al. [45] investigated the effects of stress on Mexican immigrants to the US. Their findings indicated that the level of social support that women participants derived from their family networks was negatively correlated with AL. This relationship,

however, was only apparent for women who lived in majority White communities as opposed to Mexican enclaves. Specifically, women who lived in White society and who had low family support were eight times more likely to experience everyday stress and associated AL than their high-support counterparts. These results thus complement those generated by Brooks et al. [44] in terms of connection quality and value of specifically familial social ties. However, McClure et al.'s findings also suggest that external factors such as the demography of one's social environment may play a role in this relationship. These results should be interpreted with caution, though, as they are based on an exceedingly small sample size ($N$ = 126).

Extending this research further, Seeman et al. [46–48] made similar, but also more detailed discoveries that add more texture to the link between social connectedness and AL. In their initial, two-cohort study (Cohort 1: 70-79-year-olds, Cohort 2: 58-59-year-olds), the authors assessed the link between social connectedness, gender, and stress and AL. The results indicated several gender differences in the specific stress-related physiological reactivity that contributed to overall AL. In particular, across both cohorts, cardiovascular factors (e.g. BP, cholesterol, waist/hip ratio) were the most common contributors to AL in men. However, in women, neuroendocrine components (urinary cortisol, catecholamines) represented the main drivers of AL. In other words, everyday stress appeared to physically manifest in different ways as a function of gender [48]. Further, the results indicated a range of age- and gender-related differences in the extent to which social connectedness buffered against stress and the associated AL. Results for the younger cohort indicated that some ties were more significant for one gender than the other. For instance, social relationships based on intellectual/recreational connections appeared to protect women, but not men, against high AL. On the other hand, strong maternal ties were associated with lower AL for men, but not for women. Regardless of gender, however, the overall association in the younger cohort indicated that more and more positive social ties predicted lower AL scores. By contrast, in the older cohort, a higher number of social ties and increased frequency of emotional support decreased stress and AL only in men. Finally, negative spousal relationships (characterized by criticism and/or high demands) predicted higher AL in men, but not in women. Negative relationships with children, however, were associated with higher AL for both genders [48].

Comparable results were generated in two subsequent studies by Seeman et al. [46, 47]. A two-cohort study (Cohort 1: 54-70-year-olds, Cohort 2: >70-year-olds), conducted in Taiwan, found that the quantity of social connections available to the individual correlated negatively with AL across both cohorts [46]. They also found that for younger males (in Cohort 1), having a spouse predicted lower AL. Further, for both males and females in Cohort 1, negative relationships were associated with higher AL. This latter finding in particular was further supported in Seeman et al. [47] where they found that social strain (i.e. everyday stress associated with positive vs. negative ties) was the strongest, positive predictor of AL in a sample of 32-45-year-old men and women. Indeed, while initial analyses revealed that AL was negatively associated with both the number of social ties and the level of perceived available social support, both of these relationships were rendered non-significant when controlling for social strain. Taken together, these three studies by Seeman et al. [46–48] complement those by Brooks et al. and McClure et al. by reiterating the notion that the content and quality of social connections is at least as important as their sheer number in terms of the buffer effect. These studies also align by highlighting gender and age differences in both physiological reactivity to everyday stress as well as the extent to which social connectedness buffers against such stress and AL.

The results generated by Seeman et al. [46–48] resonate strongly with more recent research, which delves deeper into the link between social connectedness, stress/AL, and CMD [38, 39, 49]. In their first study, Yang et al. [39] operationalized AL in terms of C-reactive protein

(CRP), fibrinogen, and albumin serum levels. These are all known markers of stress-related physiological inflammation and risk factors for various cancers and CVD. Reasoning that social disconnection leaves the individual more vulnerable to the effects of everyday stress, participants' physiological data was correlated with self-report measures of the extent of social network integration (SNI, measuring size of social network and frequency of contact). Complementing past research, the results indicated an overall negative association between SNI and measures of stress and AL. These correlations, however, were more pronounced for men than for women. Specifically, adjusting for age and race, men who were socially isolated were 58% more likely than their socially connected counterparts to have elevated CRP levels and 94% more likely to have fibrinogen levels in the highest quartile. By contrast, socially isolated women were 38% more likely than socially connected women to record high-risk levels of fibrinogen. They also only had slightly and non-significantly elevated CRP levels. Mapping these results onto mortality, the results showed that everyday stress and the associated AL accounted for statistically significant proportions of the association between social isolation and all-cause mortality (14%), CVD mortality (12%), and cancer mortality (24%).

Next, in a similar study using a cross-sectional design, Yang et al. [38] again found significant and negative associations between SNI and stress/AL. With SNI values categorized from 1 = low to 4 = high, results showed that individuals who scored an SNI of 3 or lower, exhibited increasingly heavier stress and AL burdens than those who scored a 4. The pattern across SNI categories thus mimicked a dose-response relationship. In particular, participants with an SNI score of 3, 2, or 0–1 were 1.49 (95%CI 1.08, 2.06), 1.69 (95%CI 1.21, 2.36), and 2.35 (95%CI 1.62, 3.40) times more likely, respectively, to have elevated AL than those who scored a 4. Further, adjusting for education level, the SNI-AL relationship observed in the total sample, was significantly more pronounced for low-educated people. Within this subsample, compared to participants who had an SNI of 4, those who scored 2–3 were twice as likely to have elevated AL, while people who scored 0–1 were three times as likely. Finally, the association between SNI and AL was also stronger in non-White than White populations. While this association was statistically non-significant, the authors speculated that this was due lack of statistical power in their non-White sample ($n$ = 157). This variation between SES levels may signify disproportionate access to other stress-buffering resources than social connectedness (e.g. high-SES individuals may have better financial security/rainy day funds to fall back on when needed).

In another two-wave (six-year) longitudinal study, Yang et al. [49] gauged AL by systolic blood pressure (SBP) and hypertension diagnoses. They used the same SNI measure as in their earlier work, but with an added social support component. Their results were again consistent with their previous research. There was an overall inverse correlation between SNI and both SBP ($p$ < 0.01) and hypertension ($p$ < 0.01). Further, individuals with the lowest level of SNI were nearly three times as likely as their high-SNI counterparts to be hypertensive (OR = 2.72, 95%CI 1.75, 4.21). The results from the longitudinal analyses showed that individuals with low social support at Wave 1 experienced a slight increase in SBP at Wave 2 ($\beta$ = 0.03, $p$ = 0.03). However, this effect became non-significant when SNI was added to the analysis. In terms of hypertension, respondents with the lowest SNI score had a 75.3% increase in risk from Wave 1 to Wave 2 (six years later). Results also suggested that this effect was partially mediated by social support, though this association was statistically non-significant.

Only a single quasi-experimental study was retained in the search results [50]. This study measured glucocorticoid sensitivity, cortisol levels, and cytokine production as AL indicators in two groups of parents: One group with a chronically ill child ($n$ = 25) versus another with a healthy child ($n$ = 25). The former parent group experienced significantly more stress than the latter. Social connectedness was assessed in terms of group membership and social support.

There were no statistically significant between-group differences in cytokine production, however parents of chronically ill children showed significantly less dexamethasone suppression of IL-6 production than parents with healthy children. In terms of cortisol levels, parents with ill children had lower morning levels ($p < .01$) than their counterparts. Group membership and instrumental social support interacted to impact negatively and significantly on glucocorticoid sensitivity in parents with ill children, but in parents with healthy ones.

Finally, seven articles identified in the literature search reported weaker or null overall associations between the principal variables of interest [51–57]. In a study focusing on the stress-buffering aspects of religious group memberships, Maselko et al. [52] ($N = 853$ 70-80-year-olds) found no statistically significant relationship between the number of social ties and AL. They did, however, discover that weekly religious service attendance was associated with a 61% decreased risk of high AL for women, but not for men. The physiological reactivity was driven mainly by epinephrine levels and waist-hip ratio. In spite of their lacking results for social ties, the authors speculated that their measure of social connection and integration was too general in scope, and thus may have failed to capture any effects of social connectedness. Similarly, Weinstein et al. [53], Hawkley et al. [51], Gruenewald et al. [56], and Friedman et al. [57] investigated the relationship between SES-related stressors and AL. While each study found negative relationships between social connectedness measures an AL, none were statistically significant, with the exception of Weinstein who reported a significant positive association between widowhood (a marker of social connectedness) and AL ($\beta = 0.24$, $p < .05$). Lastly, Gersten [54] and Glei et al. [55] conducted separate studies in Taiwan. Both papers report negative, but weak relationships between social connectedness and AL.

## Discussion

### Main findings

Taken together, the results reported in the reviewed papers support the general conclusion that having ties to other persons or groups protects the individual against everyday stress and the associated AL. Controlling for a broad range of covariates, 13 out of 20 retained studies reported significant and inverse links between social connectedness, stress, and AL. The remaining seven studies consistently reported inverse, albeit non-significant associations between these factors [51–57]. These results dovetail nicely with the extant literature on the strong positive association between social connectedness and physical health described in the introduction [4–6]. Specifically, the themes that emerge from the research further bolsters the notion that one way that social connectedness protects the individual from chronic disease is by buffering against stress and AL. Ultimately, this reiterates the fact that weak social connectedness is a major health-risk factor, and one that should be front and center in CMD- and cancer-preventive interventions, health-risk algorithms, clinical screening procedures, and the like.

While the *overall* relationship between social connectedness and AL is relatively clear, the pattern of underlying mechanisms that emerged in the review adds a layer of complexity. In particular, it was evident that gender, SES, as well as the quality and nature of the individual's social connections often decide the extent to which social connectedness strengthens or weakens individual resilience to stress and AL. Specifically, and as one might expect, a negative and/or strained relationship seems to be more of a liability than an advantage [47], while positive and supportive relationships have the opposite effect [46–48]. Furthermore, men appear to benefit more from spousal and parental ties as well as from general social connectedness than women do [39, 46, 48]. On the other hand, women seem to gain more than men from friendships and general familial relationships [45, 48]. Finally, SES markers also appeared to

moderate the relationship, with the link between social connectedness and AL being stronger for low-educated and/or minority populations [38, 45]. While the reviewed evidence base is relatively small, the magnitude of the findings is augmented in the context of past research, indicating that the well-being effects of social connectedness are moderated by a variety of factors, including quality, content (e.g. social status and gender), and nature of individuals' multiple social connections [5, 58–60].

## Strengths and limitations

In spite of its relatively small size in terms of sheer study quantity, the reviewed evidence base has a number of redeeming strengths. As indicated in the results section, the systematic quality assessment of the literature suggests that the studies discussed here are mostly of high empirical quality. Only three of 20 studies were deemed as methodologically "weak", with most of the rest coming out as "strong". These high ratings are mainly due to the rigorous research designs employed, the mostly large and representative study populations, as well as the effort to control for a broad range of potentially extraneous variables (see Table 1). In particular, the longitudinal designs employed by Brody et al., Brooks et al., Seeman et al., Weinstein et al., and Yang et al. are noteworthy as strong approaches to the measurement of the link between social connectedness and AL. Most of these studies tapped relatively large and representative populations, often across a decade or more, and as a result generated highly compelling findings and conclusions.

Another strength relates to the physiological nature of the main outcome variable. Most research on the costs and benefits of social connectedness rely on self-report scales of well-being and health, and as such is prone to the considerable bias and error associated with subjective measures of this sort. As mentioned in the introduction, this may be part of the reason why weak social connectedness, in spite of the evidence, is treated by clinicians and researchers as a second-tier health-risk factor. The reviewed studies, however, circumvent many of these validity issues of self-report measures by focusing on purely physiological outcomes of stress and their relation to social connectedness. This expounds the very real transference of negative psychological experience to physical health-risk factors.

The use of broad, multi-component AL measures in most of the studies also speaks to the level of methodological rigor employed. The pertinence of measuring multiple markers of AL is especially evident when considering the gender differences in the physiological manifestation of stress seen in Yang et al. [39]. In other words, inclusive AL measures may be necessary to capture any variation in reactivity that may exist across different populations.

There are also a few limitations that need to be addressed. First, all of the studies included in the present review were correlational in nature, and thus cannot account for any causal relationships. In the context of relevant past experimental research, however, the causal impact of social connectedness on physiological stress reactivity may be surmised with some level of probability. For instance, several laboratory paradigms have demonstrated the protective qualities of social connectedness by experimentally manipulating social support during acute stress. Typically, these studies indicated that the physical or mental presence of social support (e.g. a friend or confederate accompanying the participant; making social connectedness mentally salient) during acute stress minimized the physiological reactivity to that stress compared to no/minimal-support control groups [8, 9, 60–64].

Another limitation relates to the fact that all but three studies were based in the US, and only four studies focused on non-Western populations (Taiwanese). This is a considerable shortcoming as past research has found cultural differences (e.g. between Western individualist and Eastern collectivist culture) in resilience to stressors as well as in the nature, structure,

and perceived value of social networks and relationships [65]. Given these differences, it is conceivable that the relationship between social connectedness and AL observed in Western studies may play out differently in non-Western populations. Indeed, the Taiwanese studies were disproportionately more likely to report null associations between social connectedness and AL. Specifically, of a total four studies on Taiwanese populations, three indicated inconclusive or non-significant results. Thus, cross-cultural comparisons could potentially provide deeper insight into the social connectedness-AL relationship.

In addition, while the frequent use of multi-component AL measures represents an overall strength in the reviewed research, the cross-study variability in AL indicators may nonetheless complicate efforts to synthesize and compare the results of the different papers. For example, while some studies focused on neuroendocrine markers of AL, others focused on cardiometabolic indices. Future research might mitigate this issue by planning studies around the Allostatic Load Index (ALI) [36] where possible.

Similarly, the variation across studies in the measurement of social connectedness also represents a potential limitation to the literature. Most of the reviewed studies operationalized connectedness by number and characteristics of individual ties and relationships. However, other research indicates that conceptualizing social connection in terms of group memberships rather than interpersonal ties is a stronger predictor of health and well-being [66]. Specifically, this research indicates that the groups with which we identify (e.g. gender, ethnicity, profession, hobby, etc.) provide us with a variety of shared social identities that enable the individual to conceptualize the self in terms of their group belonging ("us" and "we") rather than merely a personal identity ("me" and "I"). Faced with adversity, this sense of group-based community and solidarity may facilitate individual resilience and psychological capital [5, 7, 60, 66–68]. Additionally, because social identities are domain specific they allow the individual to shift away from a stressful identity (e.g. one's job) and tap into other, more positive and harmonious ones for respite and support (e.g. hobby, religion) [5, 59, 60, 69]. Given these findings, future research should focus on whether multiple group membership buffers against AL in a similar fashion as interpersonal relationships do.

The issue of our broad definition of social connectedness, follows naturally from the previous point. As noted in the methods section, we deliberately accepted an operationalization of social connectedness that included various measures of, for instance, social network size, quality, and content, social support, social activity and integration, etc. Thus, the reviewed studies are somewhat lacking in terms of a clear definition of social connectedness. Nonetheless, consistent with past research, our results strongly indicate that it is both the extent and quality of social connectedness that protect the individual against AL. Hence, in terms of applying social connectedness as a screenable health-risk factor, this suggests that in addition to gauging the mere extent or presence of an individual's social network, clinical measures should also focus on the *quality* (e.g. social support, integration, activity) of that network.

Finally, as mentioned previously, the evidence base is quite small. For this reason, only the overall negative relationship between social connectedness and AL stood out with any clarity. However, this result may be skewed by publication bias favoring significant results. Further, the present review identified several potential mechanisms that may underpin the association between social connectedness and AL. These relate mainly to gender, social status, and quality and nature of social connections. However, more research drilling down into the true significance of these particular factors is needed.

## Conclusion and future directions

The present review advances the notion of weak social connectedness as a significant health-risk factor. It does so by emphasizing the psychophysiological mechanism by which social

connectedness moderates everyday stress and thus AL. The bigger and more supportive the individual's social network is, the lower the likelihood that she/he will experience AL. Understanding the underlying mechanisms of this relationship opens up promising avenues for primary-care interventions targeting CMD-risk factors. It also stresses that weak social connectedness could and should be regarded in clinical settings as a significant health-risk factor. While this review highlights the negative overall association between social connectedness and AL, the current evidence base is limited in size and scope. Firm conclusions about the veracity of moderators identified in individual studies therefore remain elusive. In the context of applying measures of social connectedness in preventive clinical settings, these shortcomings in the evidence warrant further research to totally clarify when and how social connectedness impacts on AL. Specifically, the present review highlights gender, SES, and cultural variation as potential, but under-researched moderators. The need to unravel the numerous ways of conceptualizing and measuring social connectedness are also emphasized. The authors hope that this review will serve as a jumping-off point for future research to unpack the total relationship between social connectedness, AL, and CMD.

## Supporting information

**S1 Table. PRISMA 2009 checklist.**
(DOC)

## Author Contributions

**Conceptualization:** Anders Larrabee Sonderlund, Trine Thilsing, Jens Sondergaard.

**Data curation:** Anders Larrabee Sonderlund.

**Formal analysis:** Anders Larrabee Sonderlund.

**Methodology:** Anders Larrabee Sonderlund, Trine Thilsing.

**Project administration:** Anders Larrabee Sonderlund.

**Supervision:** Trine Thilsing, Jens Sondergaard.

**Writing – original draft:** Anders Larrabee Sonderlund.

**Writing – review & editing:** Anders Larrabee Sonderlund, Trine Thilsing, Jens Sondergaard.

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
