## [Decision Letter · Decision Letter 0]

6 Nov 2019

PONE-D-19-23360

Should social disconnectedness be included in primary-care screening for cardiometabolic disease? A review of the relationship between everyday stress, social connectedness, and allostatic load

PLOS ONE

Dear Dr Larrabee Sonderlund,

Thank you for submitting your manuscript to PLOS ONE. After careful consideration, we feel that it has merit but does not fully meet PLOS ONE’s publication criteria as it currently stands. Therefore, we invite you to submit a revised version of the manuscript that addresses the points raised during the review process.

Both reviewers were quite positive, but offered some minor suggestions to improve the current manuscript. Please address these minor concerns to the best of your ability. 

We would appreciate receiving your revised manuscript by Dec 21 2019 11:59PM. To enhance the reproducibility of your results, we recommend that if applicable you deposit your laboratory protocols in protocols.io, where a protocol can be assigned its own identifier (DOI) such that it can be cited independently in the future. For instructions see: http://journals.plos.org/plosone/s/submission-guidelines#loc-laboratory-protocols

We look forward to receiving your revised manuscript.

Kind regards,

Kenzie Latham-Mintus, PhD, FGSA

Academic Editor

PLOS ONE

Journal Requirements:

2. Please modify the title to identify the work as a systematic review as required by the PRISMA checklist

3. Please amend your manuscript to include your abstract after the title page.

4. Please include your tables as part of your main manuscript and remove the individual files. Please note that supplementary tables (should remain/ be uploaded) as separate "supporting information" files.

Reviewers' comments:

Reviewer's Responses to Questions

**Comments to the Author**

1. Is the manuscript technically sound, and do the data support the conclusions?

Reviewer #1: Yes

Reviewer #2: Yes

2. Has the statistical analysis been performed appropriately and rigorously? 

Reviewer #1: Yes

Reviewer #2: Yes

3. Have the authors made all data underlying the findings in their manuscript fully available?

Reviewer #1: Yes

Reviewer #2: Yes

4. Is the manuscript presented in an intelligible fashion and written in standard English?

Reviewer #1: Yes

Reviewer #2: Yes

5. Review Comments to the Author

Reviewer #1: This manuscript presents a review of 20 studies concerning the stress-buffering effect of social connection. Despite heterogeneous measurement approaches regarding aspects of social relationships, the majority of studies reviewed found that positive and supportive social relationships are associated with reduced allostatic load. In view of this, the authors make the case that a measure of ‘social connectedness’ would have some utility when screening for risk of cardiometabolic diseases, alongside the more widely-used lifestyle factors.

Overall it looks like the review was done with care, and the manuscript is presented very clearly - I enjoyed reading it. However, my main concern is that the concept of ‘social connectedness’ is left poorly-defined. The authors keep the concept intentionally broad for the purposes of the review, but they do not then go on to distill it down to a concrete, workable definition that could be used for screening purposes. As the authors acknowledge, the stress-buffering effects of social relationships appear to be a matter of quality over quantity – clearly, not all social relationships are necessarily supportive, and indeed some relationships can themselves be source of stress. The implication, presumably, would be that administering a social support scale would be more appropriate than asking people how big their social network is?

Overall I think the paper would make a good contribution to the literature, but the authors could refine their recommendations by, for example, devoting more attention to which measurement approaches tended to yield stronger effects.

Reviewer #2: I really enjoyed reviewing this paper as it is an interesting area of research with huge implications for health. It was very well-written and easy to follow, I thought the methodology was very rigorous and followed PRISMA guidelines, and all of the conclusions were well-supported, both by past literature & from findings of the review itself. I have only one minor revision suggested (see comments below) and recommend this manuscript for publication.

1. What are the main claims of the paper and how significant are they for the discipline?

The main findings from the review are that social connectedness (i.e. having important relationship with other people or groups) is protective against the effects of everyday stressors & development of AL. As an AL researcher myself, I am well-versed on this construct and really liked the way that the authors framed the importance of stress and how the maladaptive physiology contributes to a variety of chronic diseases (not just cardiometabolic disease, which was the authors’ interest). It really has immense implications for population health and has been well-validated for decades now, but because stress is often viewed as a “psychological” problem, rather than a physiological one by clinicians, it is often overlooked until chronic disease has emerged. I think the conclusion the authors make that social connectedness should be viewed as a major health risk factor is well-supported and hopefully will help change the mentality about the effects of chronic stress when it comes to screening and primary prevention.

2. Are the claims properly placed in the context of the previous literature? Have the authors treated the literature fairly?

This paper adequately summarized past literature in this field and I think fairly assessed the findings and limitations of that body of work.

3. Do the data and analyses fully support the claims? If not, what other evidence is required?

I think the data & analyses for the most part support the claims. The one thing I would be careful about when interpreting findings across the AL literature is directly comparing findings in papers that used completely different indicators of AL in their construct measures. For example, some of the studies you discussed used neuroendocrine/immune biomarkers to represent AL, while others used cardiometabolic indices to represent AL. Ideally, an AL measure would include all of those, but that isn’t always possible, so the indices vary widely across the literature, which is a known weakness of this field. So just interpret carefully when studies are using completely different metrics to represent this stress construct. And I would mention this as a study limitation too because it could skew your interpretations when synthesizing the findings of the included studies.

4. PLOS ONE encourages authors to publish detailed protocols and algorithms as supporting information online. Do any particular methods used in the manuscript warrant such treatment? If a protocol is already provided, for example for a randomized controlled trial, are there any important deviations from it? If so, have the authors explained adequately why the deviations occurred?

The PRISMA methodology was well-explained and they reference the website if someone needs to see all of the specific steps. No suggestions here.

5. If the paper is considered unsuitable for publication in its present form, does the study itself show sufficient potential that the authors should be encouraged to resubmit a revised version?

I would recommend for publication with just a minor revision that I mentioned above about adding a study limitation (interpretation of AL indices that use different biomarkers).

6. Are original data deposited in appropriate repositories and accession/version numbers provided for genes, proteins, mutants, diseases, etc.?

Not applicable.

7. Are details of the methodology sufficient to allow the experiments to be reproduced?

Yes, methodology well-described and I believe the search could be easily reproduced with the information given.

8. Is the manuscript well organized and written clearly enough to be accessible to non-specialists?

Very well-written, great paper!

6. PLOS authors have the option to publish the peer review history of their article (what does this mean?). If published, this will include your full peer review and any attached files.

Reviewer #1: No

Reviewer #2: No

---

## [Author Response · Author response to Decision Letter 0]

10 Nov 2019

Response to reviewers

We thank the reviewers for their careful and insightful revision of our paper. Their comments have absolutely enhanced the quality and clarity of our article.

Kind regards,

Anders Larrabee Sonderlund, Trine Thilsing, and Jens Sondergaard.

Reviewer 1 comments

1. Comment: Distill the broad concept of social connectedness to a more workable definition that could be used for screening purposes.

Response: We thank the reviewer for highlighting the important issue. We fully agree that our broad conceptualization of social connectedness needs to be addressed in light of the results of our review. We have added text in the Strengths and limitations section on page 27-28, l. 511-520: “The issue of our broad definition of social connectedness, follows naturally from the previous point. As noted in the methods section, we deliberately accepted an operationalization of social connectedness that included various measures of, for instance, social network size, quality, and content, social support, social activity and integration, etc. Thus, the reviewed studies are somewhat lacking in terms of a clear definition of social connectedness. Nonetheless, consistent with past research, our results strongly indicate that it is both the extent and quality of social connectedness that protect the individual against AL. Hence, in terms of applying social connectedness as a screenable health-risk factor, this suggests that in addition to gauging the mere extent or presence of an individual’s social network, clinical measures should also focus on the quality (e.g. social support, integration, activity) of that network.”

2. Comment: Refine recommendations by devoting more attention to which measurement approaches tended to yield stronger effects.

Response: This is an excellent point and we thank the reviewer for bringing our attention to it. We have inserted text in the Strengths and limitations section on page 25, line 452-456: “In particular, the longitudinal designs employed by Brody et al., Brooks et al., Seeman et al., Weinstein et al., and Yang et al. are noteworthy as strong approaches to the measurement of the link between social connectedness and AL. Most of these studies tapped relatively large and representative populations, often across a decade or more, and as a result generated highly compelling findings and conclusions.”

Reviewer 2 comments

1. Comment: Acknowledge the fact that AL indices vary across studies in interpretation of results and as an overall limitation.

Response: We thank the reviewer for picking us up on this issue, and absolutely agree that the variability in AL measurement ought to be appropriately acknowledged. We have added text in the Strengths and limitations section on page 27, line 492-496: “In addition, while the frequent use of multi-component AL measures represents an overall strength in the reviewed research, the cross-study variability in AL indicators may nonetheless complicate efforts to synthesize and compare the results of the different papers. For example, while some studies focused on neuroendocrine markers of AL, others focused on cardiometabolic indices. Future research might mitigate this issue by planning studies around the Allostatic Load Index (ALI) (36) where possible.”

---

## [Decision Letter · Decision Letter 1]

5 Dec 2019

Should social disconnectedness be included in primary-care screening for cardiometabolic disease? A systematic review of the relationship between everyday stress, social connectedness, and allostatic load

PONE-D-19-23360R1

Dear Dr. Larrabee Sonderlund,

We are pleased to inform you that your manuscript has been judged scientifically suitable for publication and will be formally accepted for publication once it complies with all outstanding technical requirements.

With kind regards,

Kenzie Latham-Mintus, PhD, FGSA

Academic Editor

PLOS ONE

Additional Editor Comments (optional):

Reviewers' comments:

Reviewer's Responses to Questions

**Comments to the Author**

1. If the authors have adequately addressed your comments raised in a previous round of review and you feel that this manuscript is now acceptable for publication, you may indicate that here to bypass the “Comments to the Author” section, enter your conflict of interest statement in the “Confidential to Editor” section, and submit your "Accept" recommendation.

Reviewer #1: All comments have been addressed

Reviewer #2: All comments have been addressed

2. Is the manuscript technically sound, and do the data support the conclusions?

Reviewer #1: Yes

Reviewer #2: Yes

3. Has the statistical analysis been performed appropriately and rigorously? 

Reviewer #1: Yes

Reviewer #2: Yes

4. Have the authors made all data underlying the findings in their manuscript fully available?

Reviewer #1: Yes

Reviewer #2: Yes

5. Is the manuscript presented in an intelligible fashion and written in standard English?

Reviewer #1: Yes

Reviewer #2: Yes

6. Review Comments to the Author

Reviewer #1: Thank you for the opportunity to review this revised submission. I am satisfied with the authors' response and have no further comments to raise. I believe the paper will make an interesting contribution.

Reviewer #2: (No Response)

7. PLOS authors have the option to publish the peer review history of their article (what does this mean?). If published, this will include your full peer review and any attached files.

Reviewer #1: No

Reviewer #2: No

---

## [Editor Report · Acceptance letter]

11 Dec 2019

PONE-D-19-23360R1 

Should social disconnectedness be included in primary-care screening for cardiometabolic disease? A systematic review of the relationship between everyday stress, social connectedness, and allostatic load 

Dear Dr. Larrabee Sonderlund:

I am pleased to inform you that your manuscript has been deemed suitable for publication in PLOS ONE. Congratulations! Your manuscript is now with our production department. 

With kind regards,

on behalf of

Dr. Kenzie Latham-Mintus 

Academic Editor

PLOS ONE